# Evaluating Data Sharing of SARS-CoV-2 Genomes for Molecular Epidemiology across the COVID-19 Pandemic

**DOI:** 10.3390/v15020560

**Published:** 2023-02-17

**Authors:** Teresa Rito, Pedro Fernandes, Raquel Duarte, Pedro Soares

**Affiliations:** 1Centre of Molecular and Environmental Biology (CBMA), Department of Biology, University of Minho, 4710-057 Braga, Portugal; 2Institute of Science and Innovation for Bio-Sustainability (IB-S), University of Minho, 4710-057 Braga, Portugal; 3Epidemiology Research Unit (EPIUnit), Institute of Public Health, University of Porto, 4050-600 Porto, Portugal; 4Laboratory of Health Community, Department of Population Studies, School of Medicine and Biomedical Sciences (ICBAS), University of Porto, 4050-313 Porto, Portugal; 5Department of Pulmonology, Hospital Center Vila Nova de Gaia/Espinho, 4500-328 Vila Nova de Gaia, Portugal; 6Clinical Research Unit of the Northern Regional Health Administration, 4000-477 Porto, Portugal

**Keywords:** genomics, SARS-CoV-2 molecular epidemiology, genetic variants, worldwide collaboration, COVID-19, phylogeography

## Abstract

Following the emergence of COVID-19 in December 2019, caused by the coronavirus SARS-CoV-2, the disease spread dramatically worldwide. The use of genomics to trace the dissemination of the virus and the identification of novel variants was essential in defining measures for containing the disease. We aim to evaluate the global effort to genomically characterize the circulating lineages of SARS-CoV-2, considering the data deposited in GISAID, the major platform for data sharing in a massive worldwide collaborative undertaking. We contextualize data for nearly three years (January 2020–October 2022) for the major contributing countries, percentage of characterized isolates and time for data processing in the context of the global pandemic. Within this collaborative effort, we also evaluated the early detection of seven major SARS-CoV-2 lineages, G, GR, GH, GK, GV, GRY and GRA. While Europe and the USA, following an initial period, showed positive results across time in terms of cases sequenced and time for data deposition, this effort is heterogeneous worldwide. Given the current immunization the major threat is the appearance of variants that evade the acquired immunity. In that scenario, the monitoring of those hypothetical variants will still play an essential role.

## 1. Introduction

The novel coronavirus SARS-CoV-2 emerged from a zoonotic transmission that most likely occurred in the region of Wuhan in China towards the end of 2019, rapidly leading to a global pandemic. SARS-CoV-2 causes severe acute respiratory disease in humans, known as coronavirus disease 2019 (COVID-19), but a wide range of symptoms have been described with COVID-19, concerning important systemic consequences, that involve major organs. This includes gastrointestinal and neurological manifestations [1,2] that often also lead to persistent symptoms that prevail long after the viral stage of the disease [3,4]. The virus spread dramatically and, in a couple of months, reached worldwide coverage and was detected in over 200 countries. Currently, over 650 million people have been diagnosed with COVID-19, with over 6.5 million deaths already caused by the disease (as of 23 December 2022; source: World Health Organization). By the end of 2020 and early 2021, several vaccines were developed providing a putative solution for the termination of the pandemic [5,6]. While these vaccines displayed varied success in terms of providing long-term immunity and efficiency against variants, they allowed for the diminishment of the global effects of the pandemic. By 13 December 2022, WHO reported over 13 billion doses of vaccines administered globally.

The early sequencing of the virus revealed a positive single-stranded RNA beta-coronavirus with a genome 29,903 nucleotides long [7] in close evolutionary proximity to coronaviruses found in bats that could represent the source of the zoonotic transmission [8,9]. Soon, thousands of genomes of SARS-CoV-2 were produced by the various countries affected. In that context, a platform became the standard deposit for data sharing of the genomes, GISAID (www.gisaid.org accessed on 30 December 2022) [10]. Currently, over 14 million genomes are deposited in GISAID, making the genome of SARS-CoV-2 the most sequenced genome in history.

The sequencing of SARS-CoV-2 genomes has been crucial in the fight against COVID-19 in relation to various aspects of national public health planning and action in each country. For example, regarding actions focusing on global scenarios, the data sharing of genomes and genotypes from the different countries has become essential in establishing policies for defining travel bans [11], confinement measures and border closures. On the national level, genetic data is an important tool when applied to trace transmission within borders [12], to identify super-spreading events through molecular epidemiology [13], to test the level of efficiency of the vaccination on local spreading [14] and even to propose measures leading to disease eradication [15]. On a global level, complete genomics applied to the populations’ genetics and phylogeography enabled the understanding of different stages of the pandemic and also enabled people to follow the spread of the virus locally and worldwide [16,17]. 

More important in the current scenario of the pandemic is the identification of new variants that could display higher virulence, transmissibility or signs indicating the potential overcoming of the immunity acquired at the individual and population levels [18] and that could cancel the advances obtained through the current vaccination efforts and natural immunization of the populations. This could ultimately revert the easing of safety and containment measures that have been achieved on a global scale. The role of molecular epidemiology has been essential in the early identification of variants such as the B.1.1.7 that emerged in the United Kingdom by the end of Summer 2020 [19] and spread globally, the highly transmittable variant B.1.1.248 that emerged in Brazil by the end of 2020, spreading to various countries and being thought to potentially evade the immune system [20], and the South African B.1.351 variant detected in October 2020 that spread globally [21].

The identification of novel variants and the tracing of the rapid spread of lineages at the national and international levels depend on the national efforts of molecular characterization through genomic sequencing of SARS-CoV-2 lineages, but also on the international data-sharing policies that have been promoted mainly through the correct deposit of those sequences on the GISAID platform [10]. It is also essential that that deposit is timely, allowing for the information on the emergence and circulation of variants to be punctually used by public health agents. For example, in the EU, the ECDC coordinates this effort, including data submission, to their “The European Surveillance System” (TESSy). Still, for global data and even data from European countries outside the EU, the GISAID website is the standard data sharing platform.

We aim to evaluate the process of international data sharing of SARS-CoV-2 genomes that has been taking place through the GISAID platform for over 30 months of the pandemic. This analysis considers the amount of data submitted, time elapsed between deposit following collection and performs comparisons between continents and countries by considering the background of incidence of COVID-19 and the general pandemic situation in each location. We will also trace the early appearance of the major clades, G, GR, GV, GK, GH, GRY and GRA, and relate how their early detection can be related with this sequencing effort.

## 2. Materials and Methods

On the 3 November 2022, we assessed the GISAID platform (www.gisaid.org accessed on 30 December 2022) from where we extracted the full deposited genomic data and corresponding metadata. The metadata file was processed using basic local scripts in Python and information was categorized in terms of the continent of origin of the deposit, country of origin, data of collection of the SARS-CoV-2 isolate, data of deposit of the genomic sequence and quality of the data (low or high coverage) and completeness.

We compartmentalized and contextualized the various levels of regional data both against the incidence values of the pandemic at various time frames and against the global sequencing effort and global incidence. Although all the data were treated, we focused our analyses on the most populated countries globally. This approach allows us to explore mathematically and visually different scenarios of low and high incidence globally. We opted for countries with population sizes over 10 million people. Values of incidence of COVID-19 for each country were obtained from the website Worldometer (https://www.worldometers.info/ accessed on 30 December 2022) assessed on the 10 November 2022. Different comparative measures were considered, such as:(a).Absolute values of shared SARS-CoV-2 genomes by each individual country and by continent since the beginning of the pandemic.(b).Total data shared monthly by each individual country and by continent. Data were analyzed from January 2020 until October 2022, although early months and later months in the analysis were not displayed.(c).Time elapsed between collection date of the SARS-CoV-2 isolate and deposit of the genomic data in the GISAID platform across different continents and countries across different periods of the pandemic (January 2020 until October 2022).(d).Hypothetical percentage of SARS-CoV-2 genomes were fully characterized given the incidence of COVID-19 at the local level (country) and more general level (continental) across the period in analysis.(e).A ratio between the percentage of data produced/shared by each country given the number of reported cases in that country against the corresponding percentage of genomes sequenced/shared given the global incidence of COVID-19; this measure provides a view of the relative contribution of each country in data sharing against the average world effort. Again, values are obtained for the full period (January 2020 until October 2022).(f).Estimated percentage of isolates not sequenced (or hypothetically not shared) of SARS-CoV-2 in each country against the corresponding worldwide number of isolates not characterized during each month of the COVID-19 pandemic (January 2020 until October 2022). While the virus does not seem to display a very high mutation rate [22,23,24], the fact is that each individual case represents a novel evolutionary line where probabilistic novel mutations can emerge. In that context the number of cases, either occurring locally or worldwide, represents the possibility of a novel mutation; from these, a few will be sequenced and characterized at the genomic level while the great majority will not. The value of incidence minus the sequenced genomes provides the number of lineages that could harbor undetected variants each month. The calculated index indicates each country’s worldwide percentage of putative, undetected variants.(g).We collected the early appearance of the major clades, G, GH, GV, GK, GR, GRA and GRY [25], on a global scale. We performed a spatial interpolation using the Kriging algorithm of the Surfer 8 software in order to geographically plot the first appearance of members of each clade according to the data deposited in GISAID. The Kriging algorithm interpolates the best linear unbiased prediction in geostatistics based on the Gaussian process regression. First appearances (for roughly the first 50 days) were plotted geographically. We also checked the phylogenetic positioning of these early cases using the Network software and the reduced median algorithm [26] as we have done before [16]. This also allowed us to correct for mislabeled clades in GISAID. Only genomes of a high coverage were used in the phylogenetic reconstruction. The reduced median algorithm while employed initially in human mitochondrial DNA data [26,27,28] has proved excellent in dealing with molecular epidemiological data [29,30]. The reduced median algorithm ran with the standard threshold of 2.0 with no weighting of the characters.

In general, data from December 2019 and January/February 2020 were assessed with caution as most countries were still initiating efforts for the genomic sequencing of SARS-CoV-2 and data from August till October 2020 might not represent a complete dataset as the time period between collection and deposit can often be long (as analyzed below), given that we focused our analysis between March 2020 and July 2022.

No ethical approval was required for this study, since the data employed were obtained as part of an online collaborative platform (www.gisaid.org, accessed at 3 November 2022), whose authors and laboratories, the sources of the sequences and metadata here analyzed, are fully acknowledged.

## 3. Results

### 3.1. Overview of Deposited Data in GISAID

On the 3rd of November 2022, over 13.5 million entrances of genomic SARS-CoV-2 data in the GISAID website were valid, considering the various parameters for which we filtered the data. The analysis of the metadata allowed us to characterize various aspects of the deposited data and even the process of deposition. We are fully aware that data will not be ideal for all entrances as, for example, the data might not have been placed with the proper collection date when depositing the genome. However, in general terms, the metadata should offer reliable trends in the genomic characterization efforts made by countries worldwide in the context of the SARS-CoV-2 global scenario.

We initially characterized the data in simple aspects (Figure 1), namely in terms of which countries contributed mostly to the total deposited data (Figure 1A) and the number of genomes deposited per month considering the absolute data per continent (Figure 1B).

In terms of individual countries that contributed the most to the deposited data, the United Kingdom and the United States of America are the two major ones (Figure 1A), the latter contributing to over one third of the data. In fact, the USA data display an evident increase in sequencing efforts from March 2021 and then even more prominently from July 2021 on (Figure 1B). Nevertheless, continent-wise, the total data from Europe corresponds to over 50% of the total deposited data in the GISAID website, with the United Kingdom representing over 40% of that data. Data from Asia, Africa, South America and Oceania correspond to less than 15% in total for the full period of the pandemic analyzed.

In terms of the number of shared sequences per month, it is clear that there has been a continuous increase from May 2020 until April 2021, with a major increase in sequencing efforts from July 2021. The peak in shared genomic data was January 2022, followed by December 2020. Since April 2022, there has been a decrease in global sequencing efforts, following the decrease in the number of COVID-19 infections worldwide, although there were over 0.5 million sequences deposited per month during this period. Considering that some data show a time lapse of months between collection and deposit (as we will discuss below), it is possible that the numbers for the last analyzed months might still slightly increase. For that reason, we did not include the last four months (August 2020 until October 2020).

It is also important to point out that the number of genomes sequenced/shared does not correlate with the numbers of the pandemic, as the results display a general increase in the percentage of sequenced genomes as the pandemic progressed until a certain point, followed by a decrease. The average ratio of sequenced genomes was 0.88% from February 2020–July 2020, followed by 1.17% (August 2020–January 2021), 2.9% (February 2021–July 2021), 2.38% (August 2021–January 2022) and 2.05% (February 2022–July 2022). Nevertheless, while the peak of data sharing was in the middle period indicated, the numbers continue to be very high in the latter periods. The average number of genomes deposited per month per continent follows the same trend. Throughout 2021, Europe and North America reached values of around 6–7% of sequenced cases, leading to the worldwide increase to nearly 3% mentioned above. 

None of the analyses displayed in Figure 1 consider the percentage of sequenced genomes given the incidence. To analyze the percentage number of cases sequenced per month (Figure 2), we built a heat map with values per month for the most populated countries worldwide (over 10 million of population). We should highlight that percentages per country are comparable between themselves but represent very different efforts. Countries with a lower number of cases can reach high percentages with much lower effort.

The analysis of the heat map displays several features that are important to indicate. It is evident that very few countries reached values above 10% in a stable way. In Africa, the countries that showed a higher percentage of sequenced genomes were Benin, across a few scattered months, and Senegal, mostly in the last year of the analysis. In Asia, Japan displays a continuous effort in sequencing a relatively high percentage of cases (more than 10%) until the end of 2021, which showed a decrease in the percentage of sequenced/shared genomes. Cambodia also shows some very high percentages in 2022. Countries such as India, that displayed high incidence across various periods, displayed low percentages across most of the pandemic. In Europe, a general increase in percentages is visible during 2021. The United Kingdom steadily displayed good percentages, while in the second half of the analyzed overall period, Sweden provided high values as well. Values approached 0% in many European countries across several months. In North America, Canada displayed steady percentages, and the USA provides continuously weak values that are caused by very high incidences values in this territory. Nevertheless, these values are better for the second half of the period analyzed. In South America, there are few points to highlight, with most values very low. Finally, Australia displayed the higher percentages of sequenced cases in the World until the middle of 2021, but such a trend is followed by clear lower numbers in the last period of analysis.

### 3.2. Elapsed Time between Collection of Samples and Data Deposit

Another vital aspect that can affect the early detection of trans-border genotypes refers to the time elapsed for the deposit of the genomic data following its collection and sequencing. This will dictate the time taken for the data to become available for other countries, which allows for the detection of transborder sharing. We present the evolution per month of the time elapsed between cases collected in that month and their deposit in GISAID (Figure 3). We present the average values in Africa (Figure 3A), Asia (Figure 3B), Europe (Figure 3C), North America (Figure 3D), Oceania (Figure 3E) and South America (Figure 3F), as well as the same value for the three countries in each continent with a higher number of deposited genomes in GISAID.

In the first year of the pandemic, the average time elapsed between collection and deposit went from one year until at least over 100 days across all the continental data. Europe, Asia and Oceania displayed the shortest average time elapsed times for this period of the pandemic; nevertheless, these values were between 100 and 200 days on average. For the second year of the pandemic (2021), the average time elapsed improved significantly across all geographic areas, decreasing to below 100 days worldwide, and ultimately, in the last analyzed months, average values of only about 20 days were obtained for Europe, North America and Oceania, showing clearly an efficient processing and depositing of the data.

### 3.3. Establishing Risk Scenarios for Undetected Variants

The importance of the molecular genotyping on a global scale depends on the values of reported cases of COVID-19. For example, countries with low incidence could be whole-sequencing a high percentage of SARS-CoV-2 strains but, despite the local importance of these studies, their global importance is lower, and even the probability of harboring novel/undetected variants is low. Countries with high incidence and large population sizes represent the geographical locations where novel variants are more likely to emerge; a scenario that, if conjugated with low genotyping rates, can ultimately result in an increased probability of undetected novel variants for an extended period of time.

Given this aspect, we aimed in the following two comparisons to display the indices that contextualize the individual countries in the global scenario. Basically, we estimated two indices that would highlight the relative position of each country in terms of genotyping efforts when comparing world values. Figure 4 and Figure 5 show the heat map of those measures. The first measure (Figure 4) basically shows the relative frequency of genotyped strains given the incidence in the country against the same global frequency of genotyping given the total cases reported worldwide in each month. Essentially, it highlights how many times more or less the percentage of SARS-CoV-2 isolates were sequenced (or at least shared in the GISAID platform) in a given country compared with the world average.

The second measure, given the incidence and sequenced cases in each country and worldwide, expresses through a percentage the negative contribution of each country to the overall frequency of genetically uncharacterized lineages in the world. It is the ratio between the uncharacterized lineages in a country against the uncharacterized lineages worldwide. It basically displays the countries where the risk for non-characterized novel variants was greater at any point. It is again important to point out that these analyses relate with data-sharing, and the measure does not exclude that authorities in each country might sequence a higher number of isolates that are not available in a collaborative platform; nevertheless, it is the international effort on that cooperation that we are evaluating.

There are a few countries that continuously displayed an average effort above the world average, namely Senegal, the United Kingdom and Canada. Australia again displayed a strong ratio above average until September 2021, following a decrease into lower averages. A similar trend is also seen for Japan and South Korea. Although they were not often above average before September 2021 as seen for Australia. Some countries, namely Sweden and Cambodia, show an opposite trend by rising their values in the late 2021 and 2022. Curiously, the USA displayed values that corresponded mainly to the global average. Several countries in Africa, Asia, North and South America were continuously below the average global values. In Europe, countries in the Eastern part of the continent showed that same continuous trend. This measure does give direct clues about the situation in terms of uncharacterized lineages worldwide.

In terms of the risk of hypothetical undetected variants (Figure 5), it is interesting that the values rarely relate directly with the overall effort displayed in Figure 4. For example, a country with high incidence would require a much more drastic effort to characterize a substantial number of the circulating SARS-CoV-2 lineages by doing only 10% of the lineages against 90% in the case of a high-incidence country.

In Africa, given that in general COVID-19 incidence was generally low (either realistic or reported), only South Africa display negative values throughout. In Asia, India followed by Iran and Turkey regularly displayed the higher global percentage of uncharacterized lineages. In Europe, higher values were obtained in countries that were actually performing an above average characterization, such as the United Kingdom, France and Germany. Despite the higher number of sequenced genomes during the end of 2020 and early 2021, the number of reported cases was very high, with the number of uncharacterized strains also increasing. Nevertheless, considering the sample size, it is unlikely that a variant rising in frequency would not be rapidly detected, as was the case with lineage B.1.1.7 in England [19]. Russia displayed negative values throughout the analysis. In North America, USA, given the high incidence in the country, a continuously negative trend was displayed. In South America, Argentina and Brazil showed some of the most negative values across the analyzed period of the paper. Australia, in accordance with the drop of deposited genomes in GISAID, moves from low values to display very negative values in 2022.

### 3.4. Exploring the Emergence and Detection of Seven Major Clades throughout the Pandemic

We went to explore the first detection of clades that obtained some level of global spread (Figure 6). For that, we established in each country the first deposited sequence of a given clade. In general, we only registered values during the first 50 days (in some cases, if the spread was much later than its initial emergence, we would register up to about 50 days after the spread). Some values were corrected from the phylogenetic analysis as some clades were mislabeled or were likely errors. We made a geographic interpolation of the data for clades G (Figure 6A), GH (Figure 6B), GR (Figure 6C), GK (Figure 6D), GV (Figure 6E), GRY (Figure 6F) and GRA (Figure 6G).

Furthermore, we obtained a phylogenetic reconstruction for the first days of detection of each clade (Figure 7) using the reduced median algorithm [26], an algorithm that proved to perform with high efficiency in the molecular epidemiology of infectious diseases [16,29,31]. Networks were colored according to the geographic location of the samples but also according to the time of collection reported in GISAID. Again, median networks were performed for clades G (Figure 7A), GH (Figure 7B), GR (Figure 7C), GK (Figure 7D), GV (Figure 7E), GRY (Figure 7F) and GRA (Figure 7G). Our objective is to perform a blank analysis based on the data from GISAID alone without focusing on published interpretations of the origin of the clades.

Regarding clade G, it likely emerged in China in January 2020, but it was only in its final form, following the mutation D614G [32], that it would dramatically expand in the following weeks. This early clade appeared in Germany mostly and in the ten days after appeared in the UK, China and Morocco (Figure 6A and Figure 7A). The Moroccan data seem to have many extra variants, raising questions regarding either the quality of the data or the real time of collection. The major expanding haplotype was still detected in several countries or just one or two mutations away. This suggests that the clade was detected across Europe early after its emergence.

Clade GH shows a curious dichotomy. Earliest samples were from Saudi Arabia (Figure 6B) with just one from Belgium that displayed many variants (possibly low quality or collection date mislabeled), but the samples from Saudi Arabia were already from a derived clade. Major spreading haplotypes were first detected in France (Figure 7B), making its origin possibly in Europe, but some early links seem to be missing. Shortly after its first detection, the clade is also detected in the USA and Zambia. 

For clade GR, the earliest detection is in China (single sequence) followed by the United Kingdom (Figure 6C). It was detected soon in continental Europe (Germany, Austria, Italy), but, according to GISAID data, also in Japan, Chile and eventually the USA. Phylogenetically, either China or the UK could be the source (Figure 7C) but given the high prominence of the ancestral G clade in Europe, Britain seems the likely source. Again, this calls attention to the issue that at this point, Britain was by far the country depositing the most sequences, and data from China was nearly absent.

The clade GK has early data detected in South Asia (Figure 6D). Phylogenetically, all the early branching haplotypes are from India (Figure 7D), leaving little doubt about its origin. The clade was soon detected in Indonesia, Oman, Israel, Timor-Leste, Malaysia, Switzerland, Slovakia, Slovenia, Italy, South Africa, Venezuela, Nicaragua and the USA (Figure 6D). What is curious about this clade is that it does not show common haplotypes between the regions, and even within India it shows a remarkable level of diversity (Figure 7D), suggesting that when it was detected it had already been evolving for a long period in South Asia. This relates with our estimate that India was one of the countries where new variants would go undetected for longer (Figure 5).

The GV clade probably emerged in continental Europe with early detections in Italy, Belgium and the UK (Figure 6E) with a possible early branching (single haplotype) collected in France (Figure 7E). Many haplotypes are shared across Europe, again suggesting early detection of the clade in Europe.

The GRY clade was detected in the UK, with early occurrences in the USA, Nigeria and Senegal (Figure 6F). These early clades outside the UK are mostly derived, suggesting that their collection date is dubious. Early branching suggests an origin in the UK (Figure 7F), and early major expanding haplotypes were detected in the UK in late September 2020, leaving little doubt despite the higher sampling performed in the UK at this period. The clade was unequivocally detected early in the USA, and it was only detected in continental Europe about 20 days after being detected in the UK.

Finally, the clade GRA, the currently wide-spread Omicron variant, shows early branching in South Africa (Figure 7G) despite having been detected earlier in India and the USA (Figure 6G). Again, it is curious that despite the fact it was a specific main subclade that spread more drastically (Figure 7G), upon the detection of the GRA clade in Southern Africa, the clade displayed high diversity and a probable considerable time of evolution. Again, Southern Africa displayed a high level of putative undetected clades in the second half of 2021 (Figure 5).

As a conclusion, it seems evident that the global clades, meaning clades that spread worldwide, are detected at the root haplotype level when they started in Europe with larger sampling against the detection of evolved clades when they emerged in countries displaying low sequencing efforts given their incidence values.

## 4. Discussion

The fight against the COVID-19 pandemic was never a fight to occur on the national level only, but one that required a global concentration of effort and cooperation between countries. While this statement is true for various diseases, the rapid spread and transmissibility of the disease made the international effort crucial, as observed during the past 36 months, where novel variants emerged at one geographic location and are soon detected globally. 

The molecular epidemiology of SARS-CoV-2 was, and still is, certainly a prime example of an obligatory international collaborative action. Molecular epidemiology, or genomic epidemiology more specifically in this case, was an important tool to detect not only the rapid spreading events in each territory, but also to detect fast-spreading strains worldwide, likely associated with increased transmissibility. These variants, beyond the immediate risk for public health represented by the increased transmissibility, also harbor the risk of avoiding previous immunization (either natural or though vaccination) [5,6,33]. In that scenario, it is also essential to rapidly characterize strains involved in cases of reinfection and previously fully vaccinated individuals in order to understand the variants of the strain that caused infection in the presence of antibodies [14]. In the current scenario, the molecular epidemiology of COVID-19 is still an essential tool, as novel “super-strains” [34] could emerge and partially revert the advances against the disease. Countries displaying limitations in vaccination, insufficient asymptomatic screening and lower sequencing efforts (mostly in Africa, Asia and South America, as displayed in Figure 1) present major risks for the undetected appearance of new variants.

In that sense, the deposited data in GISAID clearly highlights the increased importance of molecular epidemiology as the pandemic proceeded. From around the first half of the pandemic to the second half, the percentage of sequenced/deposited genomes increased dramatically, independently of the incidence values worldwide (to values above 2% of cases), a percentage that translated into an increase in the required resources. The absolute number of sequences of SARS-CoV-2 deposited throughout the analyzed period worldwide reflects changes in strategies rather than a correlation with number of cases (that would lead to a steady percentage). Given the described successes in identifying the presence of early variants in the SARS-CoV-2 European gene pool, as suggested below, a percentage above 2% could be deemed efficient; however, this value is far from being achieved across the entire European territory, and certainly very far on a global scale. 

European Union countries could be used as an example of established guidelines. ECDC published general guidelines for implementing genomic SARS-CoV-2 surveillance for EU/EEA member states (3 May 2021) [35]. For a number of cases above 100,000, it was suggested aiming at the sequencing of 1.5% of 100,000 to detect variants at 1%. However, general guidelines from ECDC also suggests 10% of the cases to be genomically investigated or for each state member to submit 500 SARS-CoV-2-positive cases sequenced to GISAID and to The European Surveillance System (TESSy) in each period of two weeks. Given an overall sampling of Europe, it would be efficient to detect newly arising variants across the territory. However, as pointed out above, the current value of 2% is clearly heterogeneous inside and outside the EU/EEA member states, where the United Kingdom greatly contributes to the overall value and Eastern European countries contributed with lower values (Figure 2, Figure 3, Figure 4 and Figure 5). Also, within the member states, the follow-up of the ECDC guidelines is achieved only in a fraction of states, as generally reported in TESSy. Nevertheless, our analyses indicate that clades that emerged in Europe were likely detected early in their evolution. The snapshot provided here highlights how a substantial sequencing effort and data sharing can lead to a fast detection of new variants within the current Omicron lineage.

A notorious improvement on this cooperation practice is the time elapsed between collection and deposit. The data processing until the deposit in the GISAID website clearly improved in the European context corresponds to an average of nearly 20 days in the last few months of the analyzed period. However, given the high-throughput sequencing technologies and bioinformatics’ tools available today, this value could be decreased to below half the time. In some cases, it is possible that the responsible entities include researchers that prefer to maintain the data until a publication is obtained. However, those practices do not benefit public health actions and, given the trend of the quick appearance of COVID-19 research in preprints before final publication, the retaining of data seems avoidable. The ECDC [18] also suggests genomic data to be both deposited in GISAID and TESSy one to two weeks upon collection; however, that time period was not achieved at any point during the pandemic. Timely available data can assure a rapid assessment for the spatio-temporal monitoring of newly established or known variants in order to guide public health actions. The ECDC plays an important role in coordinating this action in EU/EEA member states, that includes, beyond GISAID the deposit in the TESSy. However, for a coordinated global response, including Europe and outside the EU/EEA member states, GISAID plays an essential role.

Another point that might be worth worrying about in relation to the GISAID website is the lack of information on the type of sampling/collection. It would be important that they correspond to representative sampling of SARS-CoV-2 RT-PCR positive cases from population-based surveillance systems or, more specifically, that they indicate either way. In a small percentage of cases, the deposit metadata indicate that testing corresponds to travelers, which can prompt the assumption that other cases represent random sampling, but given the small amount of indicated records, this is most probably not the case. It is important that sampling ensures the representativeness of cases across the country. Given some examples of clades being detected already following a large period (possibly weeks) of evolution, it seems possible and even likely that sampling was not conducted randomly at the geographic and social levels in some countries.

## 5. Conclusions

We aimed to map the global effort of molecular epidemiology in relation to COVID-19 through the deposited data on GISAID. Europe in general and the USA after early 2021 were on the right route for a quick identification of variants. However, there is the need to increase the information on samples and improve the time elapsed until deposit. Given the current vaccination efforts worldwide and natural immunization, the major threat of significant COVID-19 resurgence in the long term is the emergence of variants that circumvent the acquired immunity. It is possible that selective pressures on the virus, due to extensive immunization, might probe the emergence of variants associated with the evasion of the immune response [36]. In that sense, genomic epidemiology will be essential to monitor their appearance and ultimately to drive research toward the design of new vaccines if necessary.

## Figures and Tables

**Figure 1 viruses-15-00560-f001:**
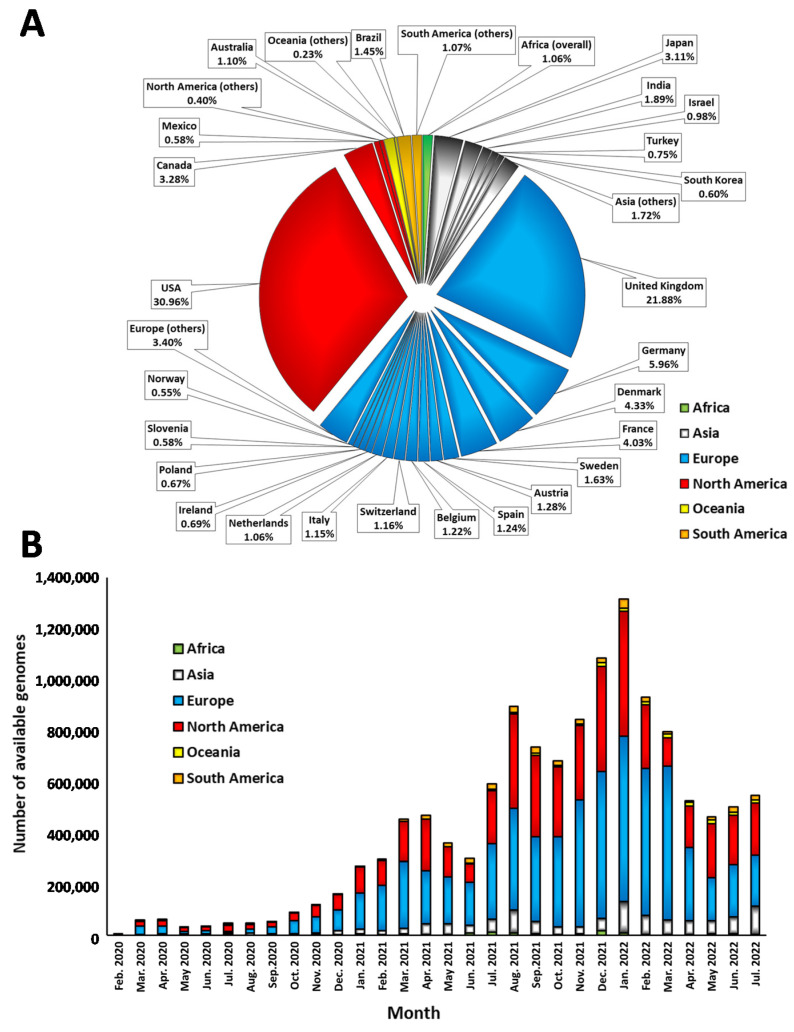
Major contributions to the deposited data in the GISAID platform in terms of submitting country (**A**) and month of submission from February 2020 until July 2022 (**B**). In both graphics, the continent of submission is indicated in the color code. In (**A**) only countries that contributed to at least 0.5% of the data are indicated individually.

**Figure 2 viruses-15-00560-f002:**
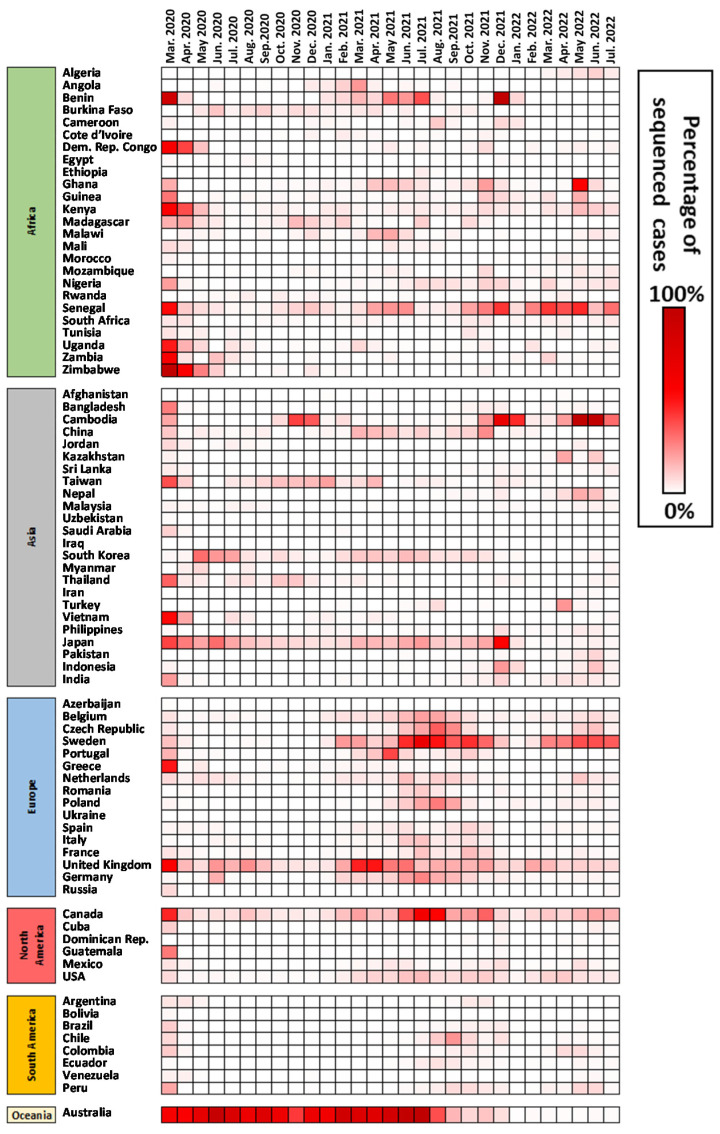
Percentage of sequenced genomes of SARS-CoV-2 given the reported incidence in different countries. Values displayed include months from March 2020 until July 2022.

**Figure 3 viruses-15-00560-f003:**
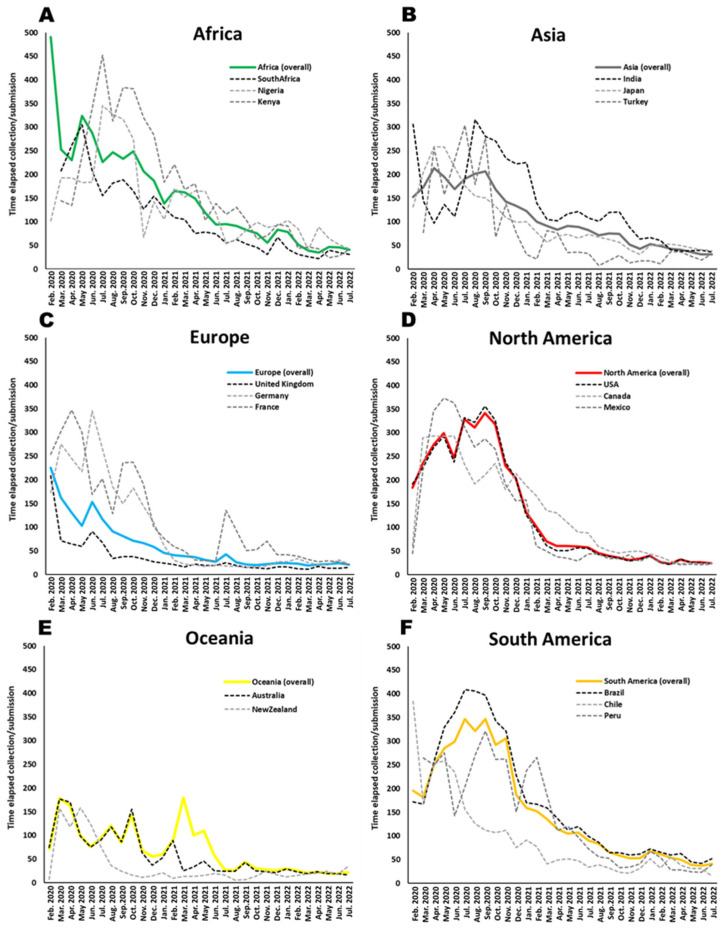
Estimated time elapsed between SARS-CoV-2 isolate collection and the deposit of the genome in the GISAID dataset for each continent; ((**A**) Africa, (**B**) Asia, (**C**) Europe, (**D**) North America, (**E**) Oceania, and (**F**) South America), between February 2020 and July 2022.

**Figure 4 viruses-15-00560-f004:**
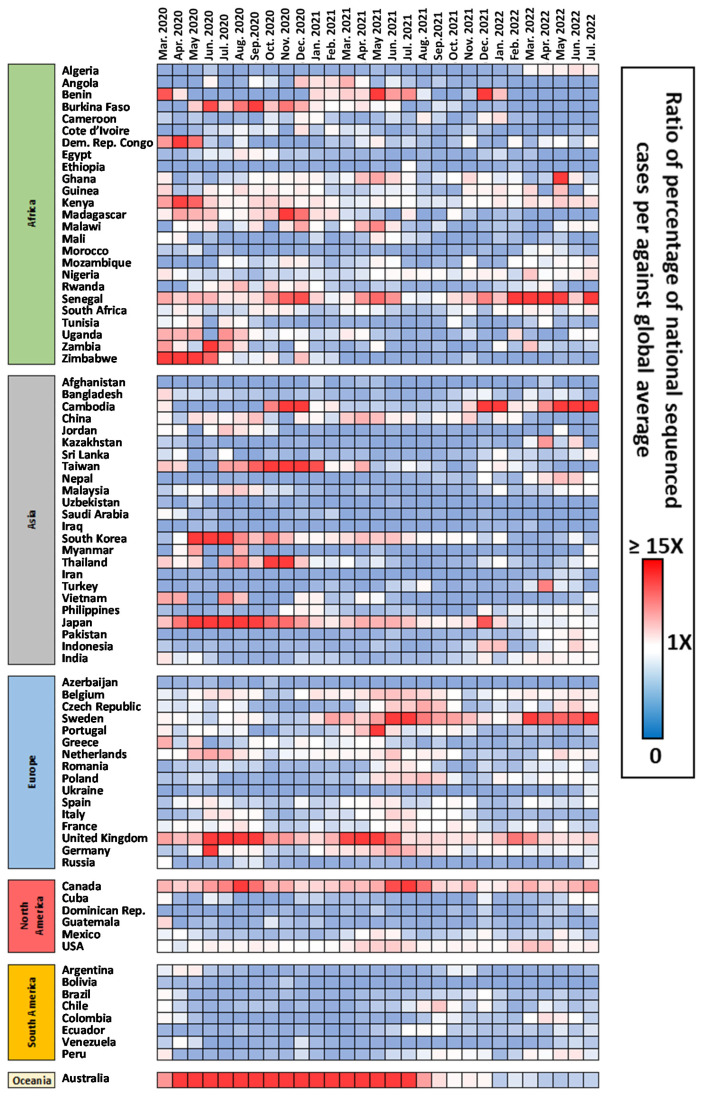
Heat map contextualizing the relative percentage of sequenced genomes in each country against the average percentages worldwide. Red points indicate that the percentage in that country/month was higher than the average worldwide, white represents a similar average, while tones of blue indicate that the average of sequenced genomes was lower than the worldwide average.

**Figure 5 viruses-15-00560-f005:**
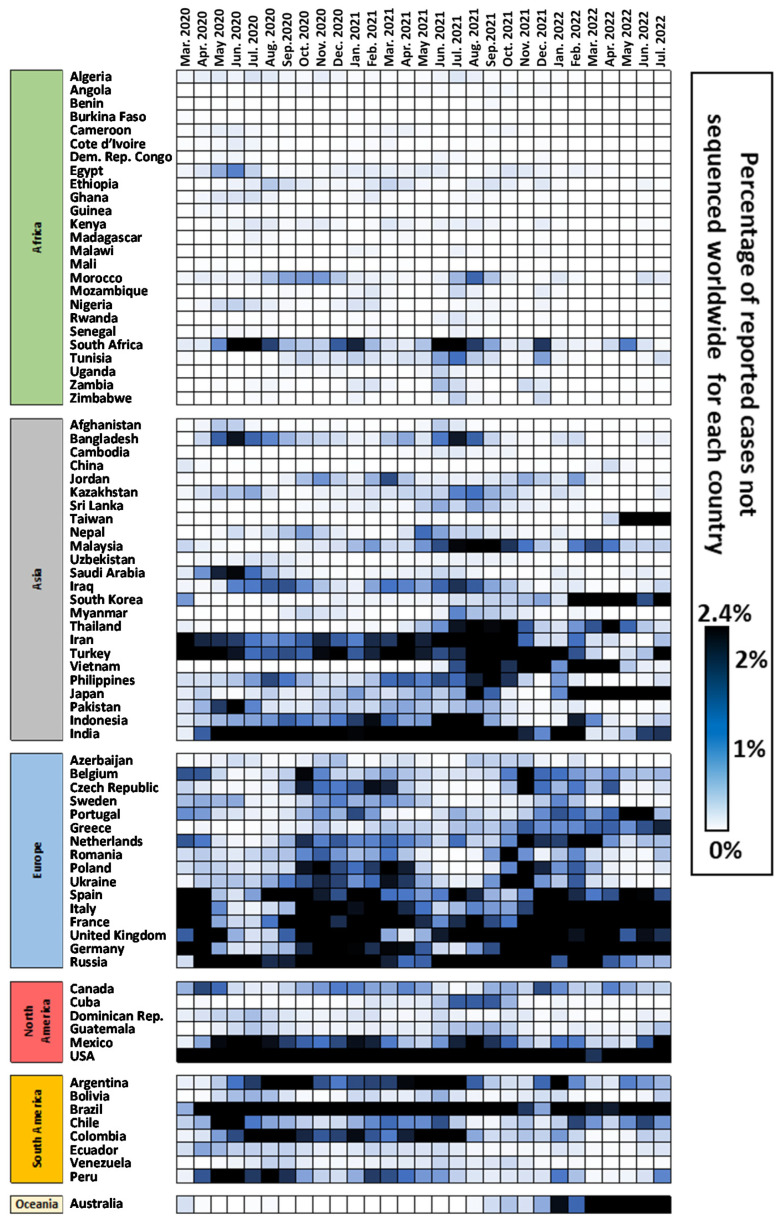
Heat map representing the percentage of hypothetical uncharacterized genomes present in each country against the hypothetical global values. A scale from white to blue to black indicates an increasing percentage. Black indicates that in that month, that country accounts for 2.4% or over of genetically uncharacterized SARS-CoV-2 genomes in the world.

**Figure 6 viruses-15-00560-f006:**
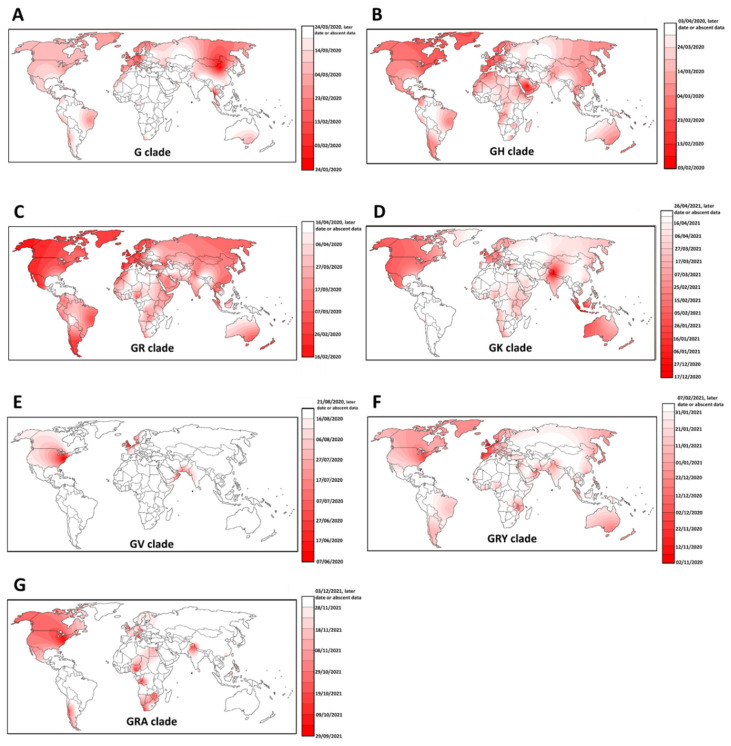
Geographic interpolation results of the first appearance of global clades as deposited in GISAID using the Kriging algorithm. Results were obtained for clades G (**A**), GH (**B**), GR (**C**), GK (**D**), GV (**E**), GRY (**F**) and GRA (**G**).

**Figure 7 viruses-15-00560-f007:**
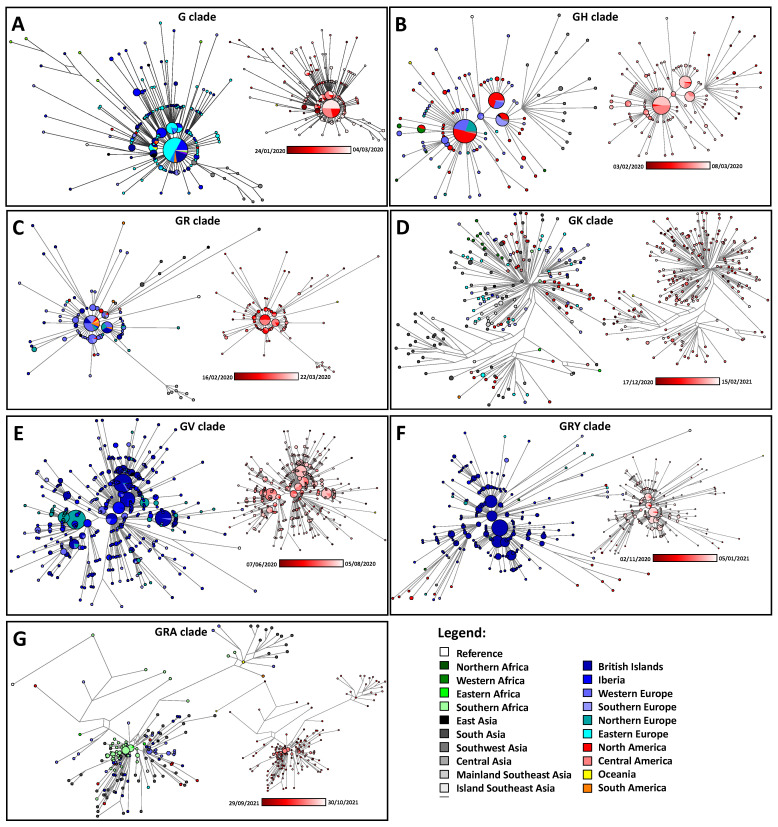
Median networks of the earliest deposited data of global SARS-CoV-2 lineages. The reference sequence was used as an outgroup in each. Networks are colored according to their geography and also according to time of collection. Networks were performed for clades G (**A**), GH (**B**), GR (**C**), GK (**D**), GV (**E**), GRY (**F**) and GRA (**G**).

## Data Availability

No new data were created or analyzed in this study.

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
