# Peer review of "Evaluating Data Sharing of SARS-CoV-2 Genomes for Molecular Epidemiology across the COVID-19 Pandemic"

_viruses, 2023, doi:10.3390/v15020560_

Round 1

Reviewer 1 Report

Authors gave a snapshot of contribution of genome deposition and data sharing of circulating SARS-CoV2 in GISAID. They also nicely presented major SARS-CoV-2 lineages. Usage of bioinformatic tools to predict the frequency of undetected and hypothetical variants are useful information. However, a few comments/concerns to improve this manuscript for ease to understand the readers are listed as following:

1.     Major reasons for new variant coming up are non-uniform availability of vaccine worldwide and insufficient asymptomatic screening.

For example: Most variants are emerging from African countries and Asian countries and Sequencing deposition is done largely from Europe and North America as shown in Figure 1. Author should comment on this issue.

2.     Highlight your analysis in case of newly emerging Omicron variant and subvariant.

3.     Describe details about the Kriging software and Surfer 8 algorithm usage to analyze the data

4.     What is correlation between number of infections of that continent in a particular month vs deposition of genome in GISAID?

Author Response

Authors gave a snapshot of contribution of genome deposition and data sharing of circulating SARS-CoV2 in GISAID. They also nicely presented major SARS-CoV-2 lineages. Usage of bioinformatic tools to predict the frequency of undetected and hypothetical variants are useful information. However, a few comments/concerns to improve this manuscript for ease to understand the readers are listed as following:

  1. Major reasons for new variant coming up are non-uniform availability of vaccine worldwide and insufficient asymptomatic screening.

For example: Most variants are emerging from African countries and Asian countries and Sequencing deposition is done largely from Europe and North America as shown in Figure 1. Author should comment on this issue.

Response: Thanks for the comment. We added in the Discussion “Countries displaying limitation in vaccination, insufficient asymptomatic screening and lower sequencing efforts (mostly in Africa, Asia and South America as displayed in Figure 1) present the major risk for the undetected appearance of new variants.”

  1. Highlight your analysis in case of newly emerging Omicron variant and subvariant.

Response: We appreciate the comment, however the manuscript relates with overall sequencing effort of SARS-CoV-2 genomes until August 2022, so the analysis of new variants of Omicron are outside the objective of the manuscript. We added in the Discussion: “The snapshot provided here highlights how a substantial sequencing effort and data sharing can lead to a fast detection of new variants within the current Omicron lineage.”

  1. Describe details about the Kriging software and Surfer 8 algorithm usage to analyze the data

Response: Thank you. We added in the methods “The Kriging algorithm interpolates the best linear unbiased prediction in geostatistics based on Gaussian process regression.”

  1. What is correlation between number of infections of that continent in a particular month vs deposition of genome in GISAID?

Response: Thanks for the comment. We added in the results “The average number of genomes deposited per month per continent follows the same trend. Throughout 2021, Europe and North America reached values of around 6-7% of sequenced cases that led to the worldwide increase to nearly 3% mentioned above.”

And in the discussion “The absolute number of sequences of SARS-CoV-2 deposited throughout the analyzed period worldwide reflects changes in strategies than a correlation with number of cases (that would lead to a steady percentage).”

Reviewer 2 Report

The authors aim to evaluate the global effort in characterizing the circulating lineages of the virus using data from GISAID, a platform for data sharing and look at the data from major contributing countries, percentage of characterized isolates, and time for data processing over nearly three years. Additionally, the author evaluates the early detection of seven major SARS-CoV-2 lineages within the collaborative effort. If authors intend to evaluate the undertaking of variant or data analysis by people, it is better to do a systematic review. Else, change the narrative of this manuscript.

The authors show in title as if they are evaluating all dat asharing platforms, in fact it i sjust GISAID, which is sometimes behind NCBI. Please edit to clarify this

The authors have attempted good work but it needs polishing before being publishable. Case sequencing and data deposition is heterogeneous worldwide. It is better to divide countries into developed, developing/middle income and lower income countries to contextualize this.

Local scripts?

details missing. language used? Scripts should be provided as supplementary.

Line 270: It is the ration? what do authors mean by this?

Heat maps should be provided in pdf for clearity. Name of countries are too small to be visible. Better is to categorize into LMIC, LIC and developed countries.

Results have no headings to divide into sections for clarity. Authors should have divided into demographic, clade-based analysis etc.

Deatils of parameters in network software in methodology for construction of Fig. 7.

Last paragraph should fall under the heading of conclusions.

Author Response

The authors aim to evaluate the global effort in characterizing the circulating lineages of the virus using data from GISAID, a platform for data sharing and look at the data from major contributing countries, percentage of characterized isolates, and time for data processing over nearly three years. Additionally, the author evaluates the early detection of seven major SARS-CoV-2 lineages within the collaborative effort. If authors intend to evaluate the undertaking of variant or data analysis by people, it is better to do a systematic review. Else, change the narrative of this manuscript.

Response: Many thanks for the comments. The objective was not to analyse the collaborative effort in terms of analysis but in terms of data sharing as often countries would have their own teams of public health agents and genetics analysts to access the current scenario. The success of fighting the COVID pandemics in terms of molecular epidemiology was the absence of required protocols for data access between nations.

The authors show in title as if they are evaluating all dat asharing platforms, in fact it i sjust GISAID, which is sometimes behind NCBI. Please edit to clarify this

Response: We thank the reviewer for the comment. GISAID was the platform that was the primal tool for public health agencies. For example European Union directives includes the data deposit in GISAID and not NCBI. We think that although GISAID is not mentioned in the title this would be the major platform for molecular epidemiology and we are clear about it in the abstract (“the data deposited in GISAID, the major platform for data sharing in a massive worldwide collaborative undertaking”)

The authors have attempted good work but it needs polishing before being publishable. Case sequencing and data deposition is heterogeneous worldwide. It is better to divide countries into developed, developing/middle income and lower income countries to contextualize this.

Response: We thank the reviewer but we must disagree. This would require a major restructuring of the manuscript and figures and we believe that it would not profit greatly the manuscript. We can point out three reasons for not using this classification: a) it changes through time and it certainly changed through the three analysed years; b) it is a classification that prolongs divisions between countries that is little related with research, policy and programming public health (see: Lencucha R, Neupane S. The use, misuse and overuse of the ‘low-income and middle-income countries’ category. BMJ Global Health 2022;7:e009067. doi:10.1136/bmjgh-2022-009067) and; c) the analysis, including the Kriging are based on geographic interpolation and it is more useful to understand geographically the problematic points of detection of variants. The focus on continent does not exclude the analysis of each individual country throughout the manuscript.

Local scripts?

details missing. language used? Scripts should be provided as supplementary.

Response: Thank you. GISAID provides the metadata as a plain file that can be easily separated into relevant partitions and extract relevant information using command line and simplistic codes that are not relevant scripts. We added in the manuscript “The metadata file was processed using basic local scripts in Python and information was categorized in terms of continent of…”

 Line 270: It is the ration? what do authors mean by this?

Response: Thanks. It is a typo. We changed the word to “ratio”.

Heat maps should be provided in pdf for clearity. Name of countries are too small to be visible. Better is to categorize into LMIC, LIC and developed countries.

Response: Thanks for pointing this out. We improved the figures. We believe now names of countries are readable. We discussed about the use of LMIC, LIC and developed countries classification above.

Results have no headings to divide into sections for clarity. Authors should have divided into demographic, clade-based analysis etc.

Response: We agree with the reviewer. We divided the results into four sub-sections:

3.1. Overview of deposited data in GISAID

3.2. Elapsed time between collection of samples and data deposit

3.3. Establishing risk scenarios for undetected variants

3.4. Exploring the emergence and detection of seven major clades throughout the pandemic

 Deatils of parameters in network software in methodology for construction of Fig. 7.

Response: Many thanks. We added in the methods: “Reduced-median algorithm ran with the standard threshold of 2.0 with no weighting of the characters.”

Last paragraph should fall under the heading of conclusions.

Response: Thanks. We added it.

Reviewer 3 Report

Authors evaluated the large data of SARS-CoV-2 genomes for molecular epidemiology in the COVID-19 pandemic. This is a nice paper. I have one minor comment: authors should further discuss about different sub-types of Omicron.

Author Response

Authors evaluated the large data of SARS-CoV-2 genomes for molecular epidemiology in the COVID-19 pandemic. This is a nice paper. I have one minor comment: authors should further discuss about different sub-types of Omicron.

Response: We thank the reviewer for the nice comment. The manuscript relates with overall sequencing effort of SARS-CoV-2 genomes until August 2022, so the analysis of new variants of Omicron are outside the established range of the manuscript. We added in the Discussion: “The snapshot provided here highlights how a substantial sequencing effort and data sharing can lead to a fast detection of new variants within the current Omicron lineage.”

Round 2

Reviewer 1 Report

They have improved data, methods and clearly stated their limitation of the manuscript in the discussion.